# Determination of Frequency-Dependent Shear Modulus of Viscoelastic Layer via a Constrained Sandwich Beam

**DOI:** 10.3390/polym14183751

**Published:** 2022-09-08

**Authors:** Ludi Kang, Chengpu Sun, Haosheng Liu, Bilong Liu

**Affiliations:** School of Mechanical & Automobile Engineering, Qingdao University of Technology, No. 777 Jialingjiang Road, Qingdao 266520, China

**Keywords:** viscoelastic material, frequency-dependent, complex shear modulus, frequency response function

## Abstract

Viscoelastic material can significantly reduce the vibration energy and radiated noise of a structure, so it is widely used in lightweight sandwich structures. The accurate and efficient determination of the frequency-dependent complex modulus of viscoelastic material is the basis for the correct analysis of the vibro-acoustic behavior of sandwich structures. Based on the behavior of a sandwich beam whose core is a viscoelastic layer, a combined theoretical and experimental study is proposed to characterize the properties of the viscoelastic layer constituting the core. In this method, the viscoelastic layer is bonded between two constraining layers. Then, a genetic algorithm is used to fit the analytical solution of the frequency¬ response function of the free–free constrained beam to the measured result, and then the frequency-dependent complex modulus is estimated for the viscoelastic layer. Moreover, by varying the length of the beams, it is possible to characterize the frequency-dependent complex modulus of the viscoelastic material over a wide frequency range. Finally, the characterized frequency-dependent complex modulus is imported into a finite element model to compute the complex natural frequencies of a sandwich beam, and a comparison of the simulated and measured results displays that the errors in the real parts are within 2.33% and the errors in the imaginary parts are within 3.31%. It is confirmed that the proposed method is feasible, accurate, and reliable. This provides essential technical support for improving the acoustic vibration characteristics of sandwich panels by introducing viscoelastic materials.

## 1. Introduction

Viscoelastic materials can effectively suppress vibration and noise in engineering structures [1,2]. However, the elastic modulus of a viscoelastic material is too small to be used as a component alone, and it is usually embedded between elastic layers to form a viscoelastic sandwich structure, which may be a sandwich beam, plate, or shell. These structures, having attractive properties in terms of strength, stiffness, lightness, and energy dissipation, are widely used in engineering fields, such as aerospace, automobile, marine and biomedical. It is of great importance to develop a method for the easy and reliable identification of the frequency-dependent shear modulus of viscoelastic materials, and this is the aim of the present work. Such a method is essential for an accurate study of the vibro-acoustic properties of a sandwich panel [3].

Nowadays, the most popular technique used to characterize the modulus of viscoelastic materials is dynamic mechanical analysis (DMA) [4,5], which allows for the measurement of a modulus depending on the frequency and temperature. Nevertheless, the experimental procedure of this technique exhibits different limits (e.g., high-frequency characterization is considerably difficult [6]) and requires expensive test equipment. Other methods, such as the creep or relaxation tests, can be used to determine the parameters of a linear viscoelastic model, but they are usually time-consuming and require strict temperature and loading control [7]. Four methods, i.e., half-bandwidth, reverberation time, power injection, and Nyquist plot, are used to measure the loss factor of a mechanical system [8]. The experimental characterization of the dynamic properties of composites is still far from standardized. Moreover, the results seem to depend on the setup because traditional methods introduce non-negligible sources of damping, such as air damping, friction at the clamps, and the mass effects of the contact excitation and measurement device [9,10]. The improvement of existing techniques or new approaches is, therefore, needed to address these drawbacks.

Over the decades, several methods have been proposed to estimate the parameters (such as Young’s modulus, shear modulus, and loss factor) of a viscoelastic material. Pritz [11] showed that the dynamic modulus obeys the same type of power law versus frequency as the loss modulus in a finite frequency range, and defined the interdependence between the dynamic modulus and loss modulus through the Kramers–Kronig dispersion relations. The validity of this method was limited within a finite bandwidth, and under these conditions, most materials exhibit a nearly constant loss factor. Bayesian approaches were proposed by Mahata et al. [12] for estimating the complex modulus of a viscoelastic material through wave propagation experiments. Bonfiglio et al. [13] determined the values of the storage and loss moduli of viscoelastic materials in a wide frequency range (100 Hz~1500 Hz) by measuring the time-domain accelerations and computing wave propagations with the transfer matrix approach. Nevertheless, this time-domain method requires a minimization procedure to determine the frequency-dependent complex modulus. To extend the work presented in Ref. [13], Bonfiglio et al. [14] presented a simplified transfer matrix approach for determining the complex modulus as a function of the frequency for homogeneous and isotropic viscoelastic materials. The setup is simplified since a top plate is not required during the test, and the complex modulus is determined directly because an analytical model is used for the measured velocity transfer function. In addition, the method allows for a narrowband measurement of the complex modulus in an extended frequency range. Adessina et al. [15] presented a finite element model based on first-order shear theory to compute the damping characteristics of sandwich structures with multi-layered frequency-dependent viscoelastic cores. Hamdaoui et al. [16] used an adjoint method to identify frequency-dependent viscoelastic damped structures. Roozen et al. [17] presented a complex wavenumber-based fitting procedure to estimate the frequency-dependent material properties of thin plates using Hankel’s functions and the image source method, and this procedure outperforms the classical approach of the spatial Fourier transform, in terms of wavenumber resolution, by a factor of 50. Wassereau et al. [18] characterized composite beams using an inverse vibratory method based on the local verification of the equation of motion applied to the Timoshenko beam. The presented method considers the composite material as a homogeneous one, and then the equivalent viscoelastic parameters can be obtained. Ablitzer et al. [19] developed an adaptation of the force analysis technique to identify the stiffness and damping properties of plates using the local equation of motion. The proposed approach is independent of the boundary conditions and may be applied at any frequency, but not necessarily a resonance. It is also valid in the mid-frequency domain, where the modal overlap is high. Pierro et al. [20] characterized the complex modulus of a viscoelastic material by fitting the measured response to an accurate analytical model based on beam dynamics, which takes into account multiple relaxation times of the material. 

Another group of feasible methods determine the parameters of a viscoelastic material by analyzing the vibration properties of construction composed of the viscoelastic material bonded between two constraining layers. Pioneering work in the analysis of the free vibration of a sandwich beam with a viscoelastic layer was conducted by Ross et al. [21], who proposed a method to determine the composite loss factor. Since then, many authors have investigated the forced vibration response of sandwich beams. Kerwin [22] firstly used the complex stiffness method to model and analyze the damping of a sandwich panel with a constrained damping layer. Mead and Markus [23] derived the sixth-order differential governing equation of a three-layer sandwich beam and studied the forced vibration of the beam using the method proposed by Ditaranto [24]. They found that boundary conditions are commonly recognized as a sensitive factor, especially in the experimental evaluation of damping properties. A general equation of the motion of a damped sandwich beam with multiple viscoelastic layers was derived by Bae et al. [25] based on the theory of Mead and Markus. 

In this paper, we present a rigorous easy-to-use approach to determine the frequency-dependent shear modulus of the viscoelastic layer constrained in a beam. A theoretical model is derived to compute the forced vibration response of a free–free constrained beam under excitation at a point. The reasons for selecting this specific system are that it is easy to set up such a freely suspended condition for conducting experiments, and the concentrated force is the most fundamental load. Based on the theoretical and experimental results of the frequency response functions (FRFs) for the constrained beam, a genetic algorithm is used to determine the frequency-dependent shear modulus of the viscoelastic material. This method can characterize the parameters over a wide frequency range by varying the length of the beam. To verify the accuracy of the frequency-dependent complex modulus determined by the proposed method, a sandwich beam consisting of the upper and lower face sheets made of aluminum (Al), a core layer of polymethacrylimide (PMI) foam and a middle layer of rubber were prepared manually for natural frequency testing. By bringing the determined modulus into the finite element model, the natural frequencies of the sandwich beam were also calculated and compared with the measured values, and the result of the comparison was satisfactory.

## 2. Theoretical Model of Forced Vibration of a Free–Free Constrained Beam

A constrained beam composed of a viscoelastic layer bonded between two constraining layers is shown in Figure 1. The thicknesses of the upper and lower constraining layers and the viscoelastic layer are $h_{1}$, $h_{3}$ and $h_{2}$, respectively; their densities are $\rho_{1}$, $\rho_{3}$ and $\rho_{2}$, respectively; the Young’s modulus of the upper and lower constraining layers are $E_{1}$ and $E_{3}$, respectively; the frequency-dependent complex shear modulus of the viscoelastic material is $G(\omega )=G_{0}(\omega )[1+i\eta (\omega )]$, where $G_{0}(\omega )$ and η(ω) are the real part and loss factor, respectively; *L* is the length and *W* is the width of the beam, and L/W≥10. The differential equation for the forced vibratory motion of a constrained beam with a viscoelastic core is given by [23]
(1)$$\frac{\partial^{6}w}{\partial x^{6}}-g(1+Y)\frac{\partial^{4}w}{\partial x^{4}}+\frac{m\omega^{2}}{D_{t}}(\frac{\partial^{2}w}{\partial x^{2}}-gw)=\frac{1}{D_{t}}\left(\frac{\partial^{2}q}{\partial x^{2}}-gq\right);$$
where the shear parameter g is
(2)$$g=\frac{G\left(\omega \right)}{h_{2}}\left(\frac{1}{E_{1}h_{1}}+\frac{1}{E_{3}h_{3}}\right);$$
the geometric parameter Y is
(3)$$Y=\frac{{\left(2h_{2}+h_{1}+h_{3}\right)}^{2}}{4D_{t}}\left(\frac{E_{1}h_{1}E_{3}h_{3}}{E_{1}h_{1}+E_{3}h_{3}}\right);$$
and the bending stiffness of the face sheets is
(4)$$D_{t}=\frac{E_{1}h_{1}^{3}}{12}+\frac{E_{3}h_{3}^{3}}{12}.$$

The transverse displacement of the constrained beam is written in the term of a modal shape function as:(5)$$w=\sum_{m=1}^{\infty }A_{m}\Phi_{m}\left(x\right)$$
where $A_{m}$ is the displacement amplitude of the *m*-th order mode. The eigenfunction describing the displacement of a free–free beam is assumed as [26]
(6)$$\Phi_{m}\left(x\right)=\cos\frac{m\pi }{L}x+\cosh\frac{m\pi }{L}x-\frac{\cos m\pi -\cosh m\pi }{\sin m\pi -\sinh m\pi }(\sin\frac{m\pi }{L}x+\sinh\frac{m\pi }{L}x)$$

Applying a point force excitation q at $x=x_{f}$:(7)$$q=F_{0}\delta (x-x_{f})$$

Inserting Equations (6) and (7) into Equation (1) yields
(8)$$\begin{array}{l} \sum_{m=1}^{\infty }A_{m}\Phi_{m}^{\left(6\right)}(x)-g(1+Y)\sum_{m=1}^{\infty }A_{m}\Phi_{m}^{\left(4\right)}(x)-\frac{m\omega^{2}}{D_{t}}\left[\sum_{m=1}^{\infty }A_{m}\Phi_{m}^{\left(2\right)}(x)-g\sum_{m=1}^{\infty }A_{m}\Phi_{m}(x)\right]\\ =\frac{1}{D_{t}}\left[-gF\delta (x-x_{f})\right] \end{array}$$

Employing the orthogonal property of eigenfunctions yields the following relation:(9)$$\frac{1}{L}\int_{0}^{L}\Phi_{m}(x)\Phi_{n}(x)dx=\delta_{mn}$$
where the delta function δ has the following property [20]:(10)$$\int_{0}^{L}F_{0}\delta (x-x_{f})\Phi_{n}(x)dx=F_{0}\delta (x_{f})$$

Both sides of Equation (9) are then multiplied by $\Phi_{n}(x)$ and integrated in [0, *L*], yielding
(11)$$A_{m}L\left\{{\left(\frac{m\pi }{L}\right)}^{6}-g(1+Y){\left(\frac{m\pi }{L}\right)}^{4}-\frac{m\omega^{2}}{D_{t}}\left[{\left(\frac{m\pi }{L}\right)}^{2}-g\right]\right\}=-\frac{g}{D_{t}}F_{0}\Phi_{m}(x_{f})$$
The displacement amplitude of the *m*-th order mode $A_{m}$ is obtained:(12)$$A_{m}=\frac{-\frac{g}{LD_{t}}F_{0}\Phi_{m}(x_{f})}{{\left(\frac{m\pi }{L}\right)}^{6}-g(1+Y){\left(\frac{m\pi }{L}\right)}^{4}-\frac{m\omega^{2}}{D_{t}}\left[{\left(\frac{m\pi }{L}\right)}^{2}-g\right]}$$
The frequency response function (FRF) can be defined as
(13)$$\begin{array}{l} H(x,x_{f},\omega )=\frac{a\left(x\right)}{F_{0}}\\ \;\;\;\;\;=\omega^{2}\frac{g}{LD_{t}}\sum_{m=1}^{\infty }\frac{\Phi_{m}(x_{f})}{{\left(\frac{m\pi }{L}\right)}^{6}-g(1+Y){\left(\frac{m\pi }{L}\right)}^{4}-\frac{m\omega^{2}}{D_{t}}\left[{\left(\frac{m\pi }{L}\right)}^{2}-g\right]}\Phi_{m}(x) \end{array}$$
where the acceleration is $a\left(x\right)={\left(j\omega \right)}^{2}w(x)$.

## 3. Sample Preparation and Experimental Setup

Two constrained beams with a viscoelastic core were prepared manually, and both possessed the same material parameters as shown in Table 1, where *B* is the width of the beams. The beam layers were bonded together by an adhesive. The surfaces of the rubber layer had been roughened before all the surfaces were cleaned and dried for easier bonding. After bonding, the constrained beams were placed between stiff plates and pressed evenly with 12 clamps along the length direction for 24 h, as shown in Figure 2a,b. The prepared constrained beams are shown in Figure 2c.

The setup used to measure the FRF of the constrained beam with a viscoelastic layer is shown in Figure 3a, and the two beams with different lengths are shown in Figure 3b. The suspension of the tested beam was made through a thin nylon wire in order to be as close as possible to the free–free boundary conditions. Impact excitation was applied at the beam section “*x_f_* = 0.4 × *L*”, and the acceleration was acquired at “*x* = 0.6 × *L*”. The model of the hammer’s tip was 086C01 (PCB), and the sensitivity was 50 mV/lbf. The model of the acceleration was 352C33 (PCB), and the sensitivity was 10.29 mV/m/s^2^. The frequency response between the hammer and the accelerometer was tested directly by the FFT module within PULSE, which was provided by B&K. The average mode of the FFT module was set to ‘Peak’. Excitation perpendicular to the surface of the constrained beam was applied with a force hammer, and the excitation point was along the centerline of the constrained beam to exclude the twisting motion. A PULSE signal acquisition system was used to simultaneously acquire and process two input signals: an acceleration signal and a force signal. 

## 4. Determination of Viscoelastic Parameters

The theoretical FRF “$H_{th}(x,x_{f},\omega )$” defined in Equation (13) has two unknown parameters in the expression of viscoelastic modulus $G(\omega )=G_{0}(\omega )[1+i\eta (\omega )]$. They are determined by fitting the theoretical FRF to the experimental FRF “$H_{exp}(x,x_{f},\omega )$”. The convergence rate of this mixed theoretical¬–experimental identification process and the resulting residual errors depend directly on the effectiveness of the optimization step, so the minimization algorithm has to be selected carefully. 

In solving combinatorial optimization problems, especially multi-objective parametric optimization problems, the genetic algorithm (GA) is usually able to obtain fast optimization results compared to some conventional optimization algorithms. To ensure the robustness, validity, and accuracy of the inverse technique, in this section we use the genetic algorithm “GA” command in the optimization algorithm toolbox of MATLAB to find the minimum value of the objective function for determining the complex modulus of the viscoelastic layer. The objective function is defined as:(14)y=∑i=1n[(Re[Hexp(x,xf,ω)]−Re[Hth(x,xf,ω)])2+(Im[Hexp(x,xf,ω)]−Im[Hth(x,xf,ω)])2]n=1,2,3,…,n
where Re[Hexp(x,xf,ω)] and Re[Hth(x,xf,ω)] are the real parts of the theoretical and experimental FRF, respectively. Im[Hexp(x,xf,ω)] and Im[Hth(x,xf,ω)] are the imaginary parts of the theoretical FRF and experimental FRF, respectively. The number of variables is two, and $\left[G_{0}\left(\omega \right),\eta (\omega )\right]$ is the target variable vector. The bounds of the variables are
(15)$$\{\begin{array}{c} \begin{array}{cc} G_{0}(\omega )\in [\begin{array}{cc} 1.0\times 10^{6} & 1.0\times 10^{9} \end{array}] & \mathrm{mm} \end{array}\\ \begin{array}{cc} \eta (\omega )\in [\begin{array}{cc} 1 & 50 \end{array}] & \% \end{array} \end{array}$$

It is worth noting that the target variable varies with the frequency; however, if the optimization is calculated directly based on the experimental data in the whole frequency range, the optimization result obtained is a constant. The processing method in this paper is to divide the tested FRFs into five groups of data and obtain an estimate based on each group of data, and the corresponding frequency is the central frequency of each group.

Figure 4a,b show the estimated results of $G_{0}\left(\omega \right)$ and η(ω), respectively, for the viscoelastic material in the constrained beam with a length of “*L* = 1.5 m”. The scattered points are the estimation directly from the grouped samples, and the red solid line is the fitted curve. From 10 Hz to 250 Hz, the scattered points are compactly distributed around the fitted curve, while from 250 Hz to 500 Hz, the dispersion of the scattered points increases. Interestingly, all the points are still distributed around the fitted curve. The results show that the proposed approach proves to be very suitable for the characterization of viscoelastic material.

In Figure 5, we compare the absolute values of the experimental FRF and theoretical FRF. At 27.5 Hz, 55 Hz, 67.5 Hz, 150 Hz, 207.5 Hz, and 470 Hz, the theoretical FRF accords with the experimental result very well. The presence of individual non-coincident peaks or valleys may be the result of unavoidable errors in the experiment. The most important reason is the deviation in the position of the excitation point or the position of the acceleration test point. Additionally, some other possible error sources include the influence of the mass added by the accelerometer, a low signal-to-noise ratio owing to a low excitation force impulse, or possible nonlinear behavior brought by a high excitation force impulse, and non-ideal boundary conditions if the accelerometer was close to a node point.

Similar reasonings can be made for the results obtained from the beam with a smaller length (i.e., *L* = 1.0 m). Figure 6a,b show the optimization results of $G_{0}\left(\omega \right)$ and η(ω) for the viscoelastic material in the constrained beam, respectively. The scattered points are compactly distributed around the fitted curve in the frequency range of 200 Hz~500 Hz, while the dispersion of the scattered points is relatively large at low frequencies (about 10 Hz~200 Hz), especially the estimated $G_{0}\left(\omega \right)$ of the viscoelastic material shown in Figure 6a, which may be caused by the low density of modes at lower frequencies. 

In Figure 7, a comparison is made between the absolute values of theoretical FRF and experimental FRF of the constrained beam with the length “*L* = 1.0 m”. At 77.5 Hz, 92.5 Hz, 200 Hz, 275 Hz, and 485 Hz, the theoretical FRF agrees with the experimental result very well. It can be observed that, as expected, the same resonance moves toward higher frequencies with the decrease in beam length.

From Figure 5 and Figure 7, in higher frequencies, the agreement between the theoretical and experimental FRFs is not very good. We believe that this may be due to the fact that as the frequency increases, the effect of the added mass of the accelerometer becomes more pronounced, and the accuracy of the accelerometer in collecting high-frequency signals decreases. In subsequent studies, other forms of sensors, such as optical sensing, will be considered as an alternative to conventional accelerometers to improve the accuracy of testing in the high-frequency range.

The real part of the shear modulus $G_{0}\left(\omega \right)$ and the loss factor η(ω) of the viscoelastic core are shown in Figure 8, for the beams with different lengths. The results suggest that it is reliable and robust to obtain the complex modulus of a viscoelastic material with the proposed technique. 

According to the results shown in Figure 4, Figure 5, Figure 6 and Figure 7, the parameters determined based on the longer beam (*L* = 1.5 m) are relatively accurate in the low-frequency range (from 20 Hz to 200 Hz), while those determined based on the shorter beam (*L* = 1.0 m) are relatively accurate in the high-frequency range (from 200 Hz to 500 Hz). Finally, it is determined that the frequency-dependent shear modulus and the loss factor of the viscoelastic material studied in this paper can be expressed as:(16)$$G_{0}\left(\omega \right)=\{\begin{array}{c} \begin{array}{cc} 1.082\times 10^{7}\times f^{0.276} & \begin{array}{cc} for & f<200 \end{array} \end{array}\\ 8.907\times 10^{6}\times f^{0.307}\begin{array}{cc} for & f\geq 200 \end{array} \end{array}$$
(17)$$\eta (\omega )=\{\begin{array}{c} \begin{array}{ccc} 0.3303\times \exp(-0.0181\times f)+0.0459\times \exp(6.571\times 10^{-4}\times f) & for & f<200 \end{array}\\ \begin{array}{ccc} 0.2976\times \exp(-9.638\times 10^{3}\times f)+0.0111\times \exp(2.887\times 10^{3}\times f) & for & f\geq 200 \end{array} \end{array}$$

## 5. Verification

In this section, the natural frequencies of a sandwich beam with a viscoelastic layer were calculated using the FEM based on COMSOL multiphysics, as shown in Figure 9. To ensure the quality of the meshing of thin face sheets while reducing the number of meshes, triangular unstructured meshes with a minimum size of 2.5 mm were created on the upper boundary of the upper face sheet and scanned to the lower boundary of the lower face sheet. The total number of hexahedral elements of this FEM model was 6800. The material types of all the layers were assumed to be linear elastic materials. The parameters of the viscoelastic layer were set to be the complex shear modulus determined above. The sandwich beam consisted of an upper and a lower face sheet made of aluminum (Al), the core layer was made of polymethacrylimide (PMI) foam, in which a middle layer of rubber was inserted, and the parameters of each layer are listed in Table 2. The length-to-width ratio of the beam was at least 10:1 so that the torsional effects were able to be neglected, and the length and width of the beam were set to be 1450 mm and 100 mm, respectively. A test sample of this sandwich beam was also prepared in the same way, as described in Section 3, and the cross-section of the sandwich beam is shown in Figure 10.

The setup used to measure the natural frequencies of the sandwich beam is shown in Figure 11. Unlike the test for FRF, an accelerometer was fixed to the lower end of the beam to prevent the modal nodes. Excitation perpendicular to the surface of the sandwich beam was applied with a force hammer, and the excitation point was along the centerline of the sandwich beam to exclude a twisting motion. Attention was paid to the force and striking position of the hammer to avoid losing the relevant modes. A PULSE signal acquisition system was used to acquire the acceleration signal. 

For a free–free sandwich beam, the natural frequencies obtained using the FEM, and through experimental tests, are shown in Table 3, and it can be concluded that these two methods give very consistent results, with relative errors of less than 2.33 % for the real parts and less than 3.31% for the imaginary parts for the first nine complex natural frequencies. Hence, it is verified that the proposed method is feasible, accurate, and reliable in estimating the dynamic complex modulus of viscoelastic material.

## 6. Conclusions

A combination of analytical and experimental methods was proposed for determining the complex parameters of a viscoelastic layer via sandwich beams in this paper. Based on the genetic algorithm, a frequency-dependent complex shear modulus of the viscoelastic core layer was characterized by fitting the analytical solutions of the forced vibration responses of the free–free viscoelastic sandwich beams with different lengths to the experimental results. The instrumentation utilized in our experiments was inexpensive and easy to use, consisting of an impact hammer, a suspended beam, and an accelerometer connected to a data acquisition module. The natural frequencies of the sandwich beam were also calculated by bringing the characterized viscoelastic material parameters into a finite element model, and a comparison of the simulated result with the measured result demonstrates that these two methods gave very consistent results, with relative errors of less than 2.33% for the real parts and less than 3.31% for the imaginary parts for the first nine complex natural frequencies. Hence, the feasibility, accuracy, and reliability were confirmed by the method proposed in this paper. In conclusion, the proposed method replaces the expensive DMA or other complex experimental methods and allows future studies to obtain the parameters of viscoelastic materials with a simple and easy approach. This provides essential technical support for improving the acoustic vibration characteristics of sandwich panels by introducing viscoelastic materials.

## Figures and Tables

**Figure 1 polymers-14-03751-f001:**
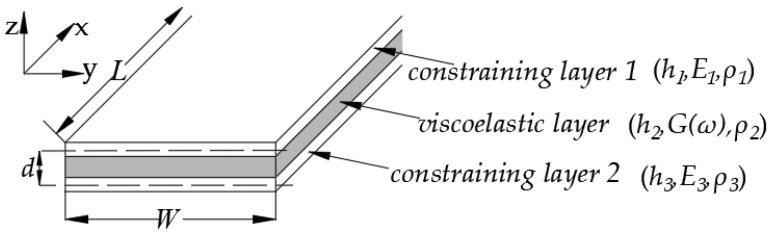
The viscoelastic material bonded between two constraining layers.

**Figure 2 polymers-14-03751-f002:**
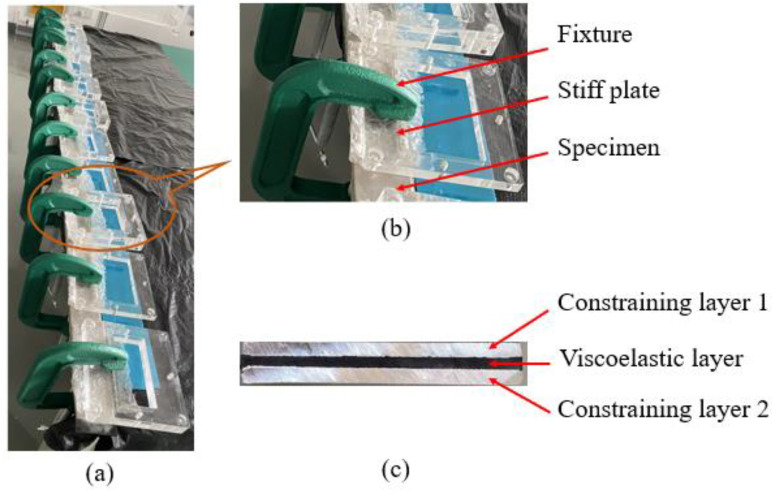
(**a**) Specimen clamping device; (**b**) detail enlargement; (**c**) the cross-section of prepared beam.

**Figure 3 polymers-14-03751-f003:**
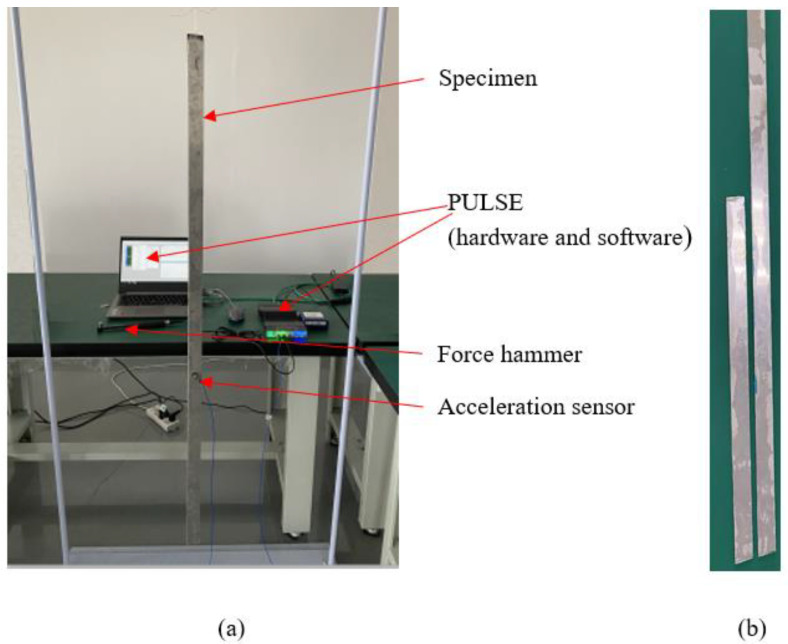
(**a**) The beam setup during the test for FRF; (**b**) the beams with different lengths.

**Figure 4 polymers-14-03751-f004:**
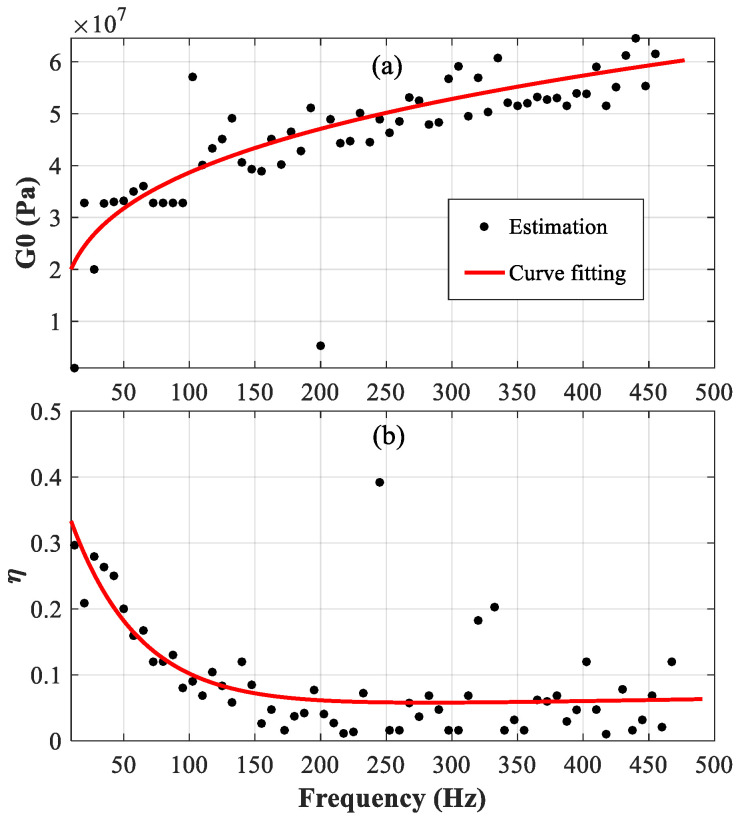
Real part of the complex shear modulus of the viscoelastic core G_0_ (ω); (**a**) the loss factor of viscoelastic core η(ω); (**b**) the constrained beam with *L* = 1.5 m.

**Figure 5 polymers-14-03751-f005:**
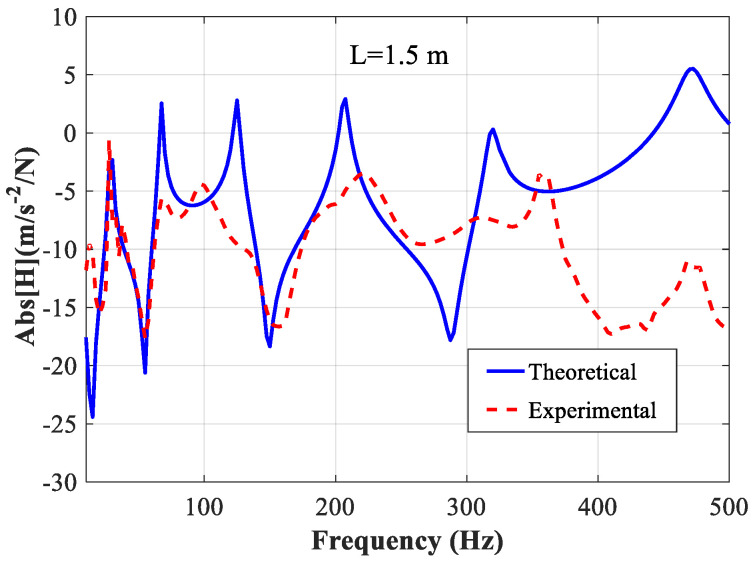
Comparison of the theoretical and experimental FRF of the beam with *L* = 1.5 m.

**Figure 6 polymers-14-03751-f006:**
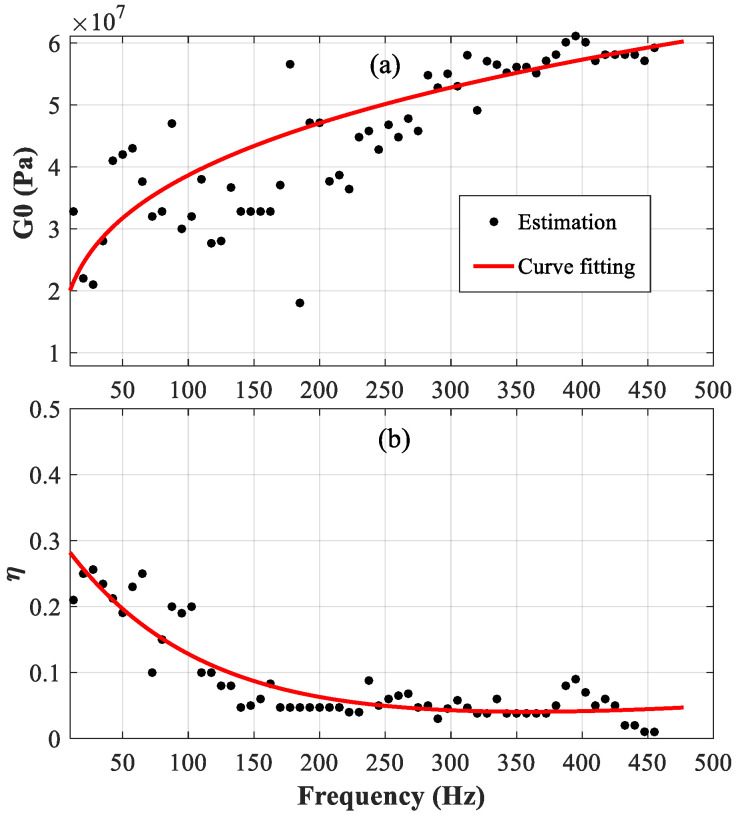
Real part of the complex shear modulus of viscoelastic core *G*_0_(*ω*); (**a**) the loss factor of viscoelastic core *η*(*ω*); (**b**) the constrained beam with *L* = 1.0 m.

**Figure 7 polymers-14-03751-f007:**
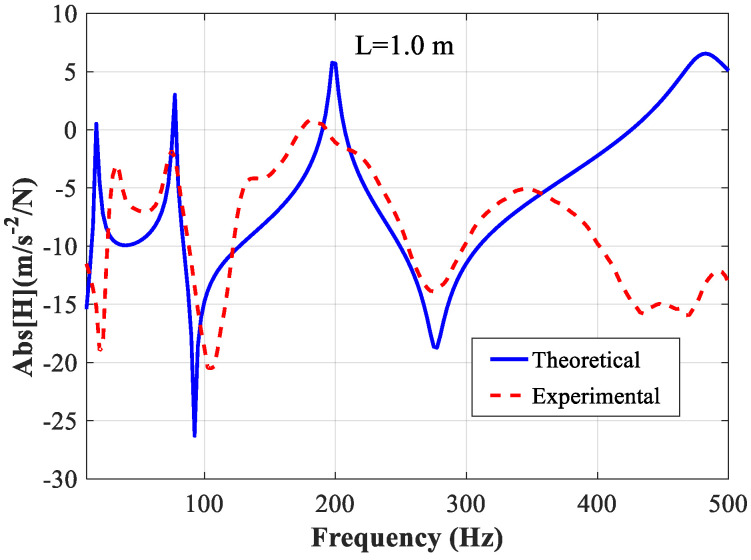
Comparison of the theoretical and experimental FRF of beam with *L* = 1.0 m.

**Figure 8 polymers-14-03751-f008:**
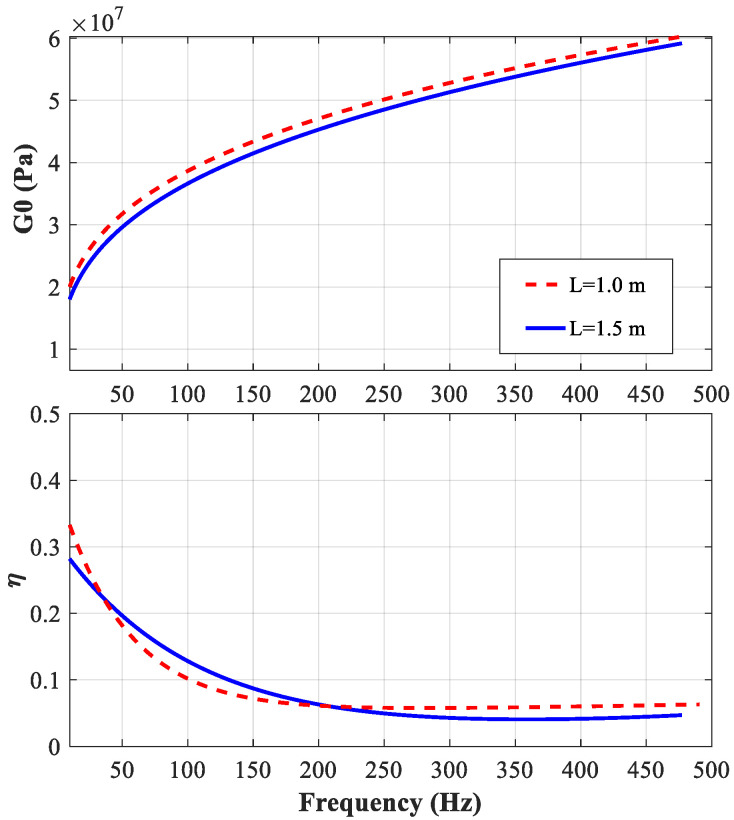
Comparisons of $G_{0}\left(\omega \right)$ and η(ω) of the viscoelastic material, obtained from the beams with different lengths.

**Figure 9 polymers-14-03751-f009:**
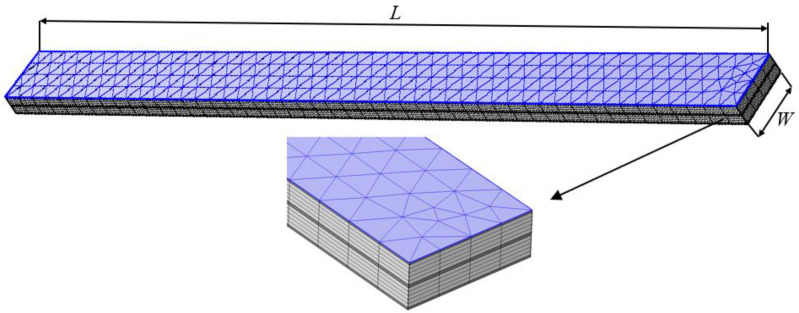
The FEM model of the beam for calculating natural frequencies.

**Figure 10 polymers-14-03751-f010:**
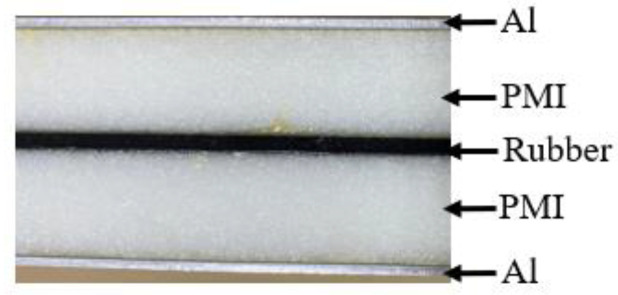
The cross-section of the beam used for the natural frequency testing.

**Figure 11 polymers-14-03751-f011:**
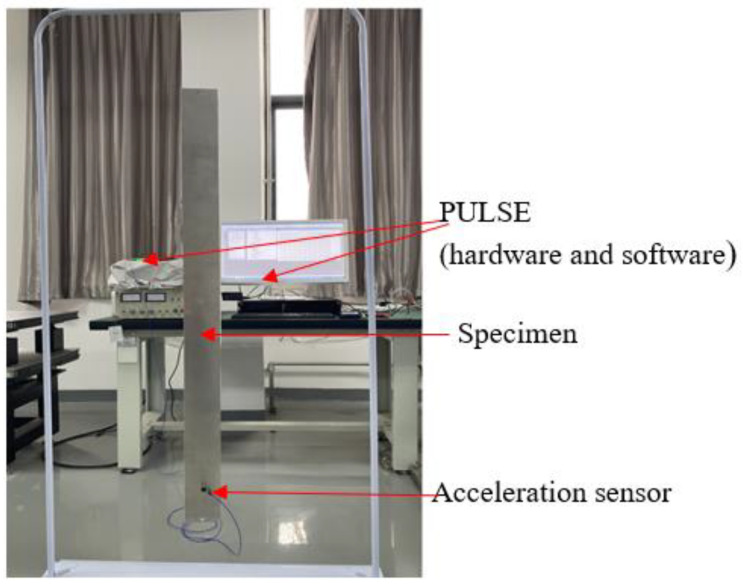
The beam setup during the test for the determination of the natural frequencies.

**Table 1 polymers-14-03751-t001:** Parameters of the viscoelastic layer and the constraining layers.

*B* (mm)	*h*_1_ (mm)	*h*_2_ (mm)	*h*_3_ (mm)	*ρ*_1_*&ρ*_3_ (kg/m^3^)	*ρ*_2_ (kg/m^3^)	*E*_1_*&E*_3_ (Pa)
40	2	2	2	2700	1100	7 × 10^10^

**Table 2 polymers-14-03751-t002:** Parameters of the sandwich beam with viscoelastic layer.

Parameters	Al	PMI	Rubber
Thickness (mm)	1.5	15	2
Density (kg/m^3^)	2700	75	1100
Young’s modulus (Pa)	70 × 10^9^	/	/
Shear modulus (Pa)	/	42 × 10^6^	*G*_0_ (*ω*)
Loss factor	0.01	0.02	*η*(*ω*)
Poisson’s ratio	0.346	0.42	0.49

**Table 3 polymers-14-03751-t003:** Predicted complex natural frequencies for laminated beams, *L* = 1450 mm, *W* = 100 mm.

Frequency	FEM (Hz)	Test (Hz)	Error of Real Part (%)	Error of Imaginary Part (%)
*f* _1_	95.6 + 3.56i	94.7 + 3.50i	0.95	1.71
*f* _2_	195.0 + 12.16i	193.3 + 11.91i	0.88	2.09
*f* _3_	295.3 + 20.4i	295.0 + 20.2i	0.10	0.99
*f* _4_	390.0 + 29.78i	392.0 + 30.3i	−0.51	−1.72
*f* _5_	485.0 + 38.23i	488.0 + 39.54i	−0.61	−3.31
*f* _6_	579.4 + 49.05i	593.2 + 50.31i	−2.33	−2.50
*f* _7_	673.6 + 56.30i	680.1 + 58.03i	−0.96	−2.98
*f* _8_	767.8 + 62.37i	776.3 + 63.50i	−1.09	−1.78
*f* _9_	863.5 + 70.96i	880.2 + 73.25i	−1.90	−3.12

## Data Availability

Not applicable.

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
