# Peer review of "Determination of Frequency-Dependent Shear Modulus of Viscoelastic Layer via a Constrained Sandwich Beam"

_polymers, 2022, doi:10.3390/polym14183751_

Round 1
Reviewer 1 Report
The Authors have subjected the manuscript entitled “ Determination of frequency-dependent shear modulus of viscoelastic layer via a constrained sandwich beam” for a possible publication in the Polymers journal.
The aim of the paper is developed a method to determine the frequency-dependent shear modulus of the viscoelastic layer constrained in a beam.
The topic is very actual and interesting. The state of the art is clearly presented. The scientific novelty is presented properly.
The paper is well organized and all the necessary information are clearly addressed.
The Reviewer has several remarks that shall be taken into account:
1. The Authors shall clearly state that the proposed method is limited to beam like structures. The analytical solution of other types of structures (plates, shells) with free boundary conditions is far more complicated if any.
2. The Authors are aware of the influence of many factors due to experimental set-up, like air damping, clamping devices, mass of the accelerometers. This shall supported by the appropriate references, e.g.:
- M. Wesolowski, E. Barkanov, Air damping influence on dynamic parameters of laminated composite plates, Measurement, Volume 85, http://dx.doi.org/10.1016/j.measurement.2016.02.036, 2016, pp. 239–248.
3. There are many research work that clearly state that the optical sensing is the most appropriate approach in the dynamic characterisation. Please justify the use of accelerometers in your research.
4. Please address the type of the hammer’s tip that was used in the experimental test. It is essential information when the hammer is used for dynamic testing.
5. Please provide the force time history and force magnitude that was used for the analytical calculations.
6. Please justify, why the bending modes are only considered in the paper. The presented beam has the width that will bring also twisting modes which will have some effect on the total displacement given by Eq. 5.
7. In the Reviewer opinion the FEM verification is not presented properly. Please consider the following remarks:
a. The FEM model shall be described in more details: number of elements, force location, ect.
b. The software used shall be addressed.
c. The material type, element type, element’s properties, ect., shall be given.
d. The FEM solution type shall be given.
e. The final model of the beam shall be presented graphically.
To sum up the paper has strong potential for the publication in Polymers journal after minor revisions. Please address the above remarks accordingly in order to proceed with the paper submission.
Reviewer 2 Report
Authors have investigated significant study related to shear modulus based on frequency model with viscoelastic layer in sandwich beam. Manuscript should be revised and need response of my following points:
· Grammatical errors should be removed.
· Abstract section should be improved by adding significant conclusions,
· General equations should be cited.
· Explanations regarding graphical section should be extended.
· Conclusion section should be improved by adding physical applications.
· Introduction section should be updated by adding following investigations
· Galerkin finite element analysis for the augmentation in thermal transport of ternary-hybrid nanoparticles by engaging non-Fourier’s law
· A dynamic assessment of various non-Newtonian models for ternary hybrid nanomaterial involving partially ionized mechanism
Reviewer 3 Report
The article deals with the identification of viscoelastic complex shear modulus via frequency response function matching for a three layered viscioelastic sandwich beam. The model of Mead and Markus is adopted and solved by means of projection on the modes of a free-free beam. For each pulsation an optimization problem is sought thanks to genetic algorithms in order to find the shear modulus and the loss factor. Then these quantities are fitted thanks to adequate functions.
The article treats an interesting subject and is well written (minor English langage reviewing is needed). But some points have to be treated before it can be published :
- The bibliography is lacking : the authors should add the following references for viscoelastic sandwich modeling and identification
https://doi.org/10.1016/j.compstruct.2016.07.056
https://doi.org/10.1016/j.jsv.2019.04.022
- The model used to build the theoretical FRF relies on assumptions : are these assumptions satisfied by the experimental setup ?
- In eq (14), the objective function is not clearly written, what does m and n stand for ?
- The proposed procedure should be tested first on numerically generated frf to validate the minimization by ga.
- The validation step proposed by the authors in §5 is incomplete, the comparison should be made on the loss factors too since the eigenfrequencies are quite note impacted by the viscoelastic modulus.
- The comparison of theoretical and experimental frf is very poor on figures 5 and 7, how the authors explain that ? does the ga minimization worked ?
- The proposed fits of the shear modulus and loss factor are built on cloud of points with high dispersion. How the authors explain that ? How the authors choosed the basis functions for the fit ?
Round 2
Reviewer 3 Report
The reviewer thanks the authors for their reply to his questions however there are still some issues to tackle before publication :
In section 5, the authors propose a verification procedure comparing COMSOL predictions of complex natural frequencies and experimental results showing relative good agreement between both approaches. However, this not the model that they are using for the identification process, they should maybe include the predictions of the Mead and Markus model for comparison. Moreover, they should compare both approaches (numerical and experimental) in terms of resonant frequencies and loss factors to ease understanding.
